# Surgical Approach to Liver Metastases in GEP-NET in a Tertiary Reference Center

**DOI:** 10.3390/cancers15072048

**Published:** 2023-03-29

**Authors:** Frederike Butz, Agata Dukaczewska, Henning Jann, Eva Maria Dobrindt, Lisa Reinhard, Georg Lurje, Johann Pratschke, Peter E. Goretzki, Wenzel Schöning, Martina T. Mogl

**Affiliations:** 1Department of Surgery, Charité—Universitätsmedizin Berlin, Corporate Member of Freie Universität Berlin, Humboldt-Universität zu Berlin, 10117 Berlin, Germany; 2Department of Gastroenterology, Charité—Universitätsmedizin Berlin, Corporate Member of Freie Universität Berlin, Humboldt-Universität zu Berlin, 10117 Berlin, Germany

**Keywords:** neuroendocrine liver metastases, debulking surgery, minimally invasive liver surgery, laparoscopic liver surgery

## Abstract

**Simple Summary:**

The choice of the surgical and therapeutic approach for patients suffering from neuroendocrine liver metastases (NELM) plays a central role in the therapeutic strategy. Whilst debulking surgery is widely accepted as an alternative approach for eligible patients, its prognostic influence remains a point of discussion. With the advent of minimally invasive liver surgery (MILS), its applicability for the treatment of neuroendocrine tumors has scarcely been described. Here, we aimed to investigate different surgical strategies in the multimodal treatment of NELM, including minimally invasive approaches. Tumor debulking showed comparable survival outcomes to curative intended liver surgery, and MILS was not inferior to open liver surgery in terms of survival rates and as such should be recommended also in patients with NELM.

**Abstract:**

Indications for liver resection in patients with gastroenteropancreatic neuroendocrine tumors (GEP-NET) vary from liver resection with curative intent to tumor debulking or tissue sampling for histopathological characterization. With increasing expertise, the number of minimally invasive liver surgeries (MILS) in GEP-NET patients has increased. However, the influence on the oncological outcome has hardly been described. The clinicopathological data of patients who underwent liver resection for hepatic metastases of GEP-NET at the Department of Surgery, Charité—Universitätsmedizin Berlin, were analyzed. Propensity score matching (PSM) was performed to compare MILS with open liver surgery (OLS). In total, 22 patients underwent liver surgery with curative intent, and 30 debulking surgeries were analyzed. Disease-free survival (DFS) was longer than progression-free survival (PFS) (10 vs. 24 months), whereas overall survival (OS) did not differ significantly (*p* = 0.588). Thirty-nine (75%) liver resections were performed as OLS, and thirteen (25%) as MILS. After PSM, a shorter length of hospital stay was found for the MILS group (14 vs. 10 d, *p* = 0.034), while neither DFS/PFS nor OS differed significantly. Both curative intended and cytoreductive resection of hepatic GEP-NET metastases achieved excellent outcomes. MILS led to a reduced length of hospital, while preserving a good oncological outcome.

## 1. Introduction

Neuroendocrine tumors (NET) are a heterogenous group of rare, relatively slow-growing malignant neoplasms that mostly arise from the gastroenteropancreatic system (GEP) [1]. Due to the usual asymptomatic nature in early stages, many patients are diagnosed at an advanced stage. Synchronous or metachronous neuroendocrine liver metastases (NELM) arise in up to 60 to 80% of all GEP-NETs [2,3]. Their occurrence has been described to be a negative predictive factor for prognosis that can impact the quality of life due to hormone-related symptoms such as diarrhea and flushing [4,5,6]. Total tumor resection in the presence of localized disease is still the only curative therapy [1]. As a complete resection of the tumor cannot be achieved in many cases, multimodal treatment strategies combining debulking surgery, local ablative and systemic therapies are recommended for advanced tumor stages [1,7]. Tumor debulking surgery for NELM has now become an established therapy approach, as a variety of studies reported favorable long-term outcome and symptom control after cytoreductive hepatic surgery [4,8,9,10,11]. In this context, clinicopathological selection criteria such as tumor grading, synchronous extrahepatic tumor manifestation and possible reduction of hepatic tumor burden are a subject of ongoing debate.

Over the last decades, minimally invasive liver surgery (MILS) has been adopted for benign and malignant hepatic tumors such as hepatocellular carcinomas and colorectal liver metastases (CRLM) [12,13]. MILS shows beneficial short-term outcomes including lower postoperative morbidity rates and shorter length of hospital stay combined with comparable oncological outcomes [12,14]. While MILS has evolved as the standard treatment for primary and secondary liver tumors, its role in NELM has only rarely been described [15].

Here, we aimed to investigate patient outcome after liver-directed surgery for NELM, dependently on the indication for surgery. Moreover, we compared outcomes after MILS to those after OLS in a sub-cohort via propensity score matching (PSM) analysis to further elucidate the role of minimally invasive liver surgery in NELM.

## 2. Materials and Methods

### 2.1. Patient Demographic and Clinical Data

Patients who underwent hepatic resection for hepatic metastases of GEP-NET at the European Neuroendocrine Tumor Society (ENETS) Centre of Excellence at the Charité—Universitätsmedizin Berlin, Germany, between January 2010 and December 2021 were identified from the Charité Comprehensive Cancer Centre (CCCC) database. Exclusion criteria were age below 18 years and primary tumor site other than the small intestine or pancreas including cancer of unknown primary (CUP). Standard demographic and clinicopathological data were collected including age, gender, American Society of Anesthesiologists (ASA) status, Body Mass Index (BMI), primary tumor site, tumor grading according to the World Health Organization (WHO) grading system and appliance of (neo-)adjuvant therapies. Additionally, surgery-related data such as indication for surgery, surgical approach, extent of resection, duration of surgery, stay in an intensive care unit (ICU), hospital stay and occurrence of complications were reviewed. Indications for hepatic resection were defined as “curative” in the case of complete resection (R0/R1) of all tumor burden, as “debulking” in the case of cytoreductive resection (R2) and as “tissue sampling” in the case of minor resection for histopathological examination (R2). Due to the non-neglectable differences in the surgical treatment of the tissue sampling group, these patients were excluded for further analyses. Minimally invasive techniques included laparoscopic and robotic-assisted surgery. Major resection was defined as resection of ≥3 adjoining liver segments according to Couinaud’s classification [16]. Postoperative complications were evaluated according to the Clavien–Dindo classification; major complications were defined as ≥3a [17]. Patients’ follow-up was performed according to the ENETS consensus guidelines [18]. Disease progression was defined via clinical assessment and cross-sectional imaging according to the Response Evaluation Criteria in Solid Tumors (RECIST).

### 2.2. Statistical Analysis

Metric variables are presented as medians (range), categorical variables as frequencies. Either the Mann–Whitney U test or the Kruskal–Wallis test was used for group comparison of continuous variables, and the chi-square test for categorical variables. The Kaplan–Meier method was used to calculate overall survival (OS), defined as the time between hepatic resection and death or the time of the last visit (loss to follow-up), disease-free survival (DFS), defined as the time between hepatic resection and the first postoperative recurrence in patients undergoing curative intended liver surgery, and progression-free survival (PFS), defined as the time between hepatic resection and the first postoperative progress. The survival rates were compared using log-rank tests.

A one-to-one propensity score matching (PSM) was performed using a logistic regression model with a match tolerance of 0.1 based on the following parameters: age, sex, ASA status, BMI, localization of the primary tumor, indication for hepatic resection, resection extent (comparing major or minor resection) and simultaneous resection of the primary tumor.

The significance level was set to 0.05. Statistical analyses were performed using SPSS Statistics software, version 27 (IBM Armonk, Armonk, NY, USA).

## 3. Results

### 3.1. Patient Characteristics

In the observed study period, a total of *n* = 66 patients met the inclusion criteria and underwent hepatic resection for hepatic metastases of GEP-NET (pNET and siNET). Of these, *n* = 14 were minor liver resections for tissue sampling and therefore were excluded from further analyses. The characteristics of the final study cohort are shown in Table 1.

In total, 29 primary tumors (56%) were located in the small intestine (siNET), whereas 23 (44%) were primary pancreatic NETs (pNET). As shown in Figure 1a,b, the grading of both primary tumor and hepatic metastases was significantly higher in pNETs than in siNETs.

In *n* = 51 (98%) cases, a resection of the primary tumor was performed, and in *n* = 23 (44%), it was simultaneous. In addition, 73% of hepatic metastases (*n* = 38) appeared synchronously.

Postoperative recurrence or progression were described in 17% (*n* = 9) and 40% (*n* = 23), respectively, of all cases, and the median time between hepatic resection and diagnosis of recurrence or progression was 24 months (3–36 months) and 10 months (2–56 months), respectively.

Within the median follow-up time of 30 months, *n* = 6 deaths (12%) were reported, and 2-year overall survival (OS) was 93% in the whole cohort (Figure 1c).

### 3.2. Outcome according to the Indication for Liver-Directed Surgery in NELM

When focusing on the indication for surgery, *n* = 22 (42%) liver surgeries were performed with curative intent, and *n* = 30 (58%) for tumor debulking. The patient characteristics are presented in Table 2.

In the curative group, all primary tumors were either low- or intermediate-grade tumors (G1 or G2 according to the WHO classification), whereas the debulking group had *n* = 1 (3%) high-grade tumor (G3).

Simultaneous resection of the primary tumor was performed in 32% and 53% of the patients in the two groups, respectively (*p* = 0.123). In addition, 37% and 30% of hepatic resections were major resections in the curative and debulking groups. The two indication groups were similar regarding the duration of surgery (*p* = 0.476), the referral to an intensive care unit (ICU) during postoperative recovery (*p* = 0.632), the length of ICU stay (*p* = 0.503), the length of hospital stay (LOS) (*p* = 0.558), 90-day complications (0.931) and 90-day major complications (0.908). In both groups, no patient died during the first 90 postoperative days.

In addition, no significant differences were found between the grading of both primary tumor and hepatic metastases. However, in the debulking group, the fractions of patients receiving neoadjuvant therapy and more than one adjuvant therapy modality were higher than in the curative group (53% vs. 27%; *p* = 0.049 and 63% vs. 32%; *p* = 0.025).

As shown in Figure 2a,b, the median disease-free survival (DFS) in the curative group was longer than the median progression-free survival (PFS) in the debulking group, i.e., 24 months vs. 10 months. However, the overall survival analysis did not reveal significant differences between the two groups, whereas the 2-year survival rates were 100% vs. 86% in the curative and debulking group, respectively (*p* = 0.588) (Figure 2c).

Neither the different localizations of the primary tumor nor the grading of hepatic metastases showed significant differences in relation to DFS/PFS and OS (Appendix A). As there was only one patient presenting with a G3 primary tumor, the G3 group was neglected for survival comparison in relation to primary tumor grading. Focusing on G1 and G2 primary tumors, DFS/PFS and OS were similar between the two groups (Appendix A).

### 3.3. Comparison of Minimally Invasive and Open Liver Surgery: Propensity Score Matching

Observing the whole cohort, *n* = 13 patients underwent minimally invasive surgery, and *n* = 39 open liver surgery. However, there were major differences with regard to the ASA status, with significantly more ASA 3 patients in the MILS group (62% vs. 31%, *p* = 0.048) (Appendix A).

As shown in Table 3, after PSM, both groups were comparable regarding sex (female: 46% vs. 46%; *p* = 1.0), median age (in years: 57 vs. 61, *p* = 0.960), median BMI (25 kg/m^2^ vs. 24 kg/m^2^; *p* = 0.801), ASA score (*p* = 0.691), localization of the primary tumor (*p* = 0.691), indication for liver resection (*p* = 1.0), resection extent (major resection: 23% vs. 38%; *p* = 0.395) and simultaneous resection of the primary tumor (31% vs. 23%; *p* = 0.658). No significant differences were found with respect to neoadjuvant and adjuvant therapy (*p* = 0.658 and *p* = 1.0), grading of the primary tumor (*p* = 0.589) and of hepatic metastases (1.0) and fraction of synchronous metastases (77% vs. 62%; *p* = 0.395), while LOS was significantly shorter in the MI group, with a median LOS of 10 days vs. 14 days (*p* = 0.034). Additionally, transfer to an intensive care unit (ICU), length of ICU stay and appearance of 90-day complications, 90-day major complications and 90-day mortality did not differ significantly between the minimally invasive and the open liver surgery groups.

The median follow-up was 25 and 30 months for MILS and OLS, respectively. As graphed in Figure 3a,b, neither DFS/PFS nor OS differed significantly between the two groups, with 2-year disease-/progression-free survival of 41% vs. 62% (*p* = 0.816), and 2-year survival rates of 92% vs. 88% (*p* = 0.392). In the MILS group, two patients died during the follow-up period due to NET-related causes, while one patient in the OLS group died.

## 4. Discussion

In patients with neuroendocrine liver metastases, liver-directed surgery plays a pivotal therapeutic role, although complete resection can be performed only in a small subset of patients. In this study, we were able to show that both curative intended and debulking surgery of NELM led to excellent outcomes. Additionally, in the here-observed cohort, our data indicated that NETs deriving from the pancreas presented with higher grading of both primary tumor and hepatic metastases compared to siNETs. Moreover, our analysis implies that in our cohort, minimally invasive liver surgery was not inferior to open liver surgery with regard to disease-/progression-free survival and overall survival, while reducing the length of hospital stay.

Therapy algorithms for patients with NELM take into account liver surgery, locoregional ablative therapies and a variety of systemic therapy regimens, whereas possible complete resection plays a central role when aiming at the curation of the disease [1]. In our cohort, 42% of NELM patients undergoing surgery were eligible for curative intended hepatectomy. The remaining 58% of the patients underwent liver resection for tumor mass reduction. Although we did not find a significant difference in overall survival between the two different indication groups throughout our follow-up period, the 2-year survival rates differed and resulted 100% for patients undergoing curative liver-directed surgery and 86% for cytoreductive surgery patients. Despite the occurrence of liver metastases, patients with NET face a relatively good prognosis compared to patients with other malignancies spread to the liver such as patients with colorectal liver metastases (CRLM). This applies in particular to patients with low-grade neuroendocrine tumors in whom noticeable effects on overall survival occur especially within follow-up on the long term. In this context, for example, Dasari et al. found that patients with distant stage-G1/G2 tumors from the small intestine and pancreas had 5-year survival rates of 69% and 50%, respectively [6]. In a different work, Selberherr et al. compared the outcomes of NELM patients undergoing surgery or staying in surveillance and demonstrated favorable outcomes for the surgery group, especially on the long term [4]. Further exploring the importance of tumor debulking in NELM patients, recently published data from a multi-center study presented reasonable long-term survival rates for patients undergoing cytoreduction for NELM, with median OS of nearly 7.5 years [8]. They presented better outcomes for patients with R0/R1 (curative resection), with 5-year OS of 85.2% compared to patients who underwent R2 (debulking) resection, who had a 5-year OS of 60.7%. Nonetheless, they demonstrated that despite the more extensive and aggressive disease, characterized by a higher incidence of lymph metastasis, a worse tumor grade and a greater liver involvement, debulking surgery offered a survival benefit for these patients. Keeping in mind the shorter follow-up period of the here-presented data, greater differences in survival between the two indication groups are likely to appear on the longer term.

Besides the question of whether or not debulking surgery is favorable in patients with non-resectable NELM, the selection criteria for patients eligible for cytoreductive therapy, especially with regard to the debulking extent, are a subject of ongoing debate [10,19]. Whereas, formerly, a debulking threshold of above 90% was aimed at, more and more authors propose to extend the debulking threshold to above 70% based on equivalent high survival rates [20,21,22]. Nonetheless, in a meta-analysis evaluating the resection extent of NELM, PFS was significantly shorter if less tumor mass could be resected [23]. In the here-presented study, the patients undergoing debulking surgery had a short PFS of only 10 months. On the other hand, when complete tumor resection was achieved, a median disease-free survival of 24 months was reached. Keeping in mind that cytoreductive surgery in NELM aims at symptom control and the amelioration of overall survival instead of a complete cure of the disease, it is not surprising that PFS was comparably short, while still preserving a favorable OS.

Interestingly, in our cohort, the patients suffering from pancreatic NET presented with higher tumor and metastatic grading than the patients with primary tumor localization in the small intestine. However, our data indicated no difference in OS or DFS/PFS between patients with primary tumors of different origins. While these results have to be interpreted with caution due to the small sample size, evidence about the association of primary tumor localization with survival and progression is inconsistent. Multiple studies promoted different survival rates for different primary tumor origins [6,24]. For example, Tierney et al. presented a longer survival for patients with NELM and a primary in the small intestine compared to patients with pancreatic, colonic, rectal and gastric primaries, whereas rectal and gastric NET patients with hepatic metastases had the shortest survival [24]. Additionally, they also showed that patients who underwent debulking surgery on the primary tumor and NELM had longer survival than patients who underwent primary tumor surgery only, surgery of metastases only or no surgery at all. In our cohort, we only included patients with hepatic resection for NELM. However, in their sub-analysis of patients undergoing debulking surgery, Tierney et al. also found that siNET patients had longer survival than pNET and colonic NET patients. On the other hand, Spolverato et al. evaluated the long-term prognosis of patients with NELM undergoing (curative intended) resection differentiating between pNET and non-pNET, including gastrointestinal and thoracic NETs, and did not find an association with the primary tumor location [3]. However, they did prove an association between extrahepatic disease and tumor grade. In the here-presented cohort, we could not reproduce a significant effect of the grading of primary tumor and hepatic metastases on patient outcome. Nonetheless, it is well established that the tumor grade affects the prognosis of GEP-NET [25,26,27]. In this regard, the small sample size and the heterogenous study population must be taken into account. Furthermore, the present study only focused on tumors with distant metastases, thus on advanced tumor stages, neglecting earlier localized tumors. Moreover, it should be underlined that, independent of primary tumor location, indication for surgery or grading, nearly all patients in our cohort (98%) had a resection of the primary, too. The effect of primary tumor resection on outcome in metastasized GEP-NET has been previously investigated, and mostly beneficial outcomes for primary-resected patients have been reported [28,29,30]. For example, Lewis et al. could prove an increase in survival, after resection of the primary tumor, of metastatic GEP-NET patients regardless of the type and extent of NELM treatment [31]. Hence, the main part of our study cohort belonged to this patient group with beneficial survival. In addition, we focused on patients who were eligible to either curative intended or cytoreductive surgery of NELM. As discussed previously, surgical therapy for NELM is a positive prognostic factor in these patients. Thus, in this selected cohort, not the tumor grade itself but rather a therapy selection bias of both primary tumor and liver metastases might have impacted the prognosis.

Experience with minimally invasive liver surgery in NELM has only scarcely been described in the literature [32,33]. To the best of our knowledge, only one study has compared MILS with OLS in NELM patients in terms of oncological outcome so far [15]. Concordant to the findings of Kandil et al., our here-presented data indicate a shorter overall hospital stay for MILS patients, while no statistical difference in either DFS/PFS or OS was observed between the two groups. In other diseases of both benign and malignant nature, the role of MILS has been widely explored, and minimally invasive procedures are now an acceptable alternative, if not the favorable choice, to OLS [12,14,34]. For example, in their PSM-based comparison of OLS and MILS in patients with colorectal liver metastases, Knitter et al. were able to confirm the efficacy of MILS in terms of oncological outcome, while preserving the advantages of minimally invasive surgery: lower postoperative complication rates, shorter length of ICU and hospital stay and lower rates of intraoperative blood transfusion [14]. Although the underlying disease cannot be compared at all, the surgical management of NELM and CRLM does show similarities: as for a relatively high risk of recurrence, tissue-sparing resections including anatomic and atypic resections are preferred [35]. Additionally, multimodal therapeutic strategies such as the preoperative induction of hypertrophy of the future liver remnant (FLR) via portal vein embolization and/or two-stage hepatectomy with clear-up of the FLR and secondary extended hepatectomy are recommended strategies for both entities [35,36]. Presumably for the non-neglectable difference in the incidence of NELM and CRLM, experience might be higher and the establishment of new and emerging surgical therapies might happen faster for CRLM compared to NELM. Therefore, with increasing expertise in minimally invasive liver surgery, an increasing role in NELM is to be expected. Based on our results, MILS can carefully be recommended for NELM patients, preserving comparable (short-term) oncological outcomes as OLS. In this regard, it should be noted that MI procedures for NELM have only been introduced over the last decades, and therefore the median follow-up for MILS patients is naturally shorter. Thus, our study highlights the importance of and provides the basis for future larger investigations to demonstrate the role of MILS in the surgical treatment of NELM patients.

Nonetheless, the current study faces some limitations. In its nature, NET are a rare and heterogenous group of tumors. Hence, our study cohort included a heterogenous population. To reduce the differences in terms of surgical treatment, we excluded patients who underwent hepatic surgery for tissue sampling and only focused on curative intended and debulking liver-directed surgery. Next, our data were collected retrospectively, and the size of our cohort was limited. Therefore, the risk of a selection bias cannot be neglected completely, and conclusions from the results, especially the comparative analyses, must be drawn cautiously. This becomes even more evident in the sub-analysis of MILS and OLS due to the relatively low number of MILS. To bypass this issue, a PSM-based comparison was performed to eliminate known confounding covariates. However, so far unknown and therefore not used confounders might have influenced our results. Additionally, one must consider that the experience of minimally invasive surgery for NELM is still limited, and as such our results may also have been influenced by the surgeon’s learning curve and a relatively short median follow-up time. Furthermore, the follow-up period of the entire cohort remained restricted to a relatively short time, especially in view of the fact that low-grade NETs progress very slowly. This means that clinically measurable effects on overall survival that might be detected on the long run, may not have been identified. Therefore, valid long-term survival outcomes cannot be confirmed by the presented study.

## 5. Conclusions

As excellent oncological outcomes could be achieved via curative intended as well as debulking surgery, hepatic resection can be recommended in NELM patients. In this regard, MILS was not inferior to OLS concerning survival rates, while reducing the length of hospital stay, and therefore offers a valid alternative to open procedures, also in NELM patients.

## Figures and Tables

**Figure 1 cancers-15-02048-f001:**
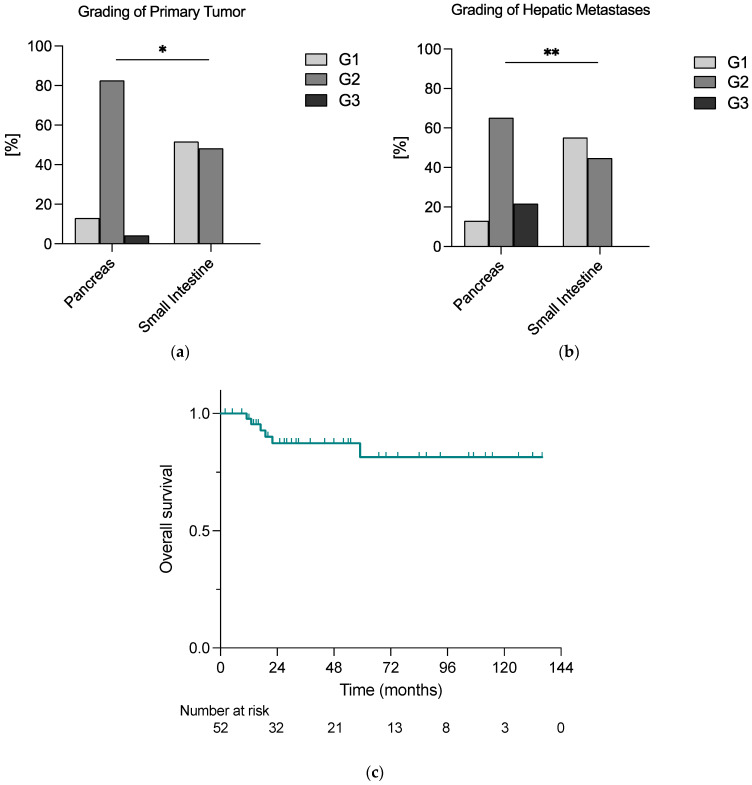
The grading of (**a**) primary tumor and (**b**) hepatic metastases differed significantly according to the localization of the primary tumor; (**c**) patients undergoing liver-directed surgery for NELM faced a relatively good prognosis, with an exemplary 2-year overall survival rate of 93% calculated with the Kaplan–Meier method; * = 0.05 ≥ *p* ≥ 0.01; ** = 0.01 > *p* ≥ 0.001.

**Figure 2 cancers-15-02048-f002:**
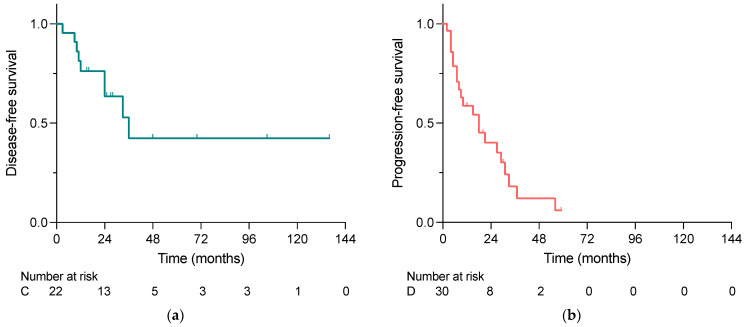
(**a**) Disease-free survival in patients undergoing curative intended liver-directed surgery for NELM, (**b**) progression-free survival in patients undergoing debulking surgery for NELM and (**c**) overall survival of patients who underwent liver-directed surgery for NELM comparing the indications for surgery calculated with the Kaplan–Meier method. No significant difference was demonstrated in OS (*p* = 0.588); C, Curative; D, Debulking.

**Figure 3 cancers-15-02048-f003:**
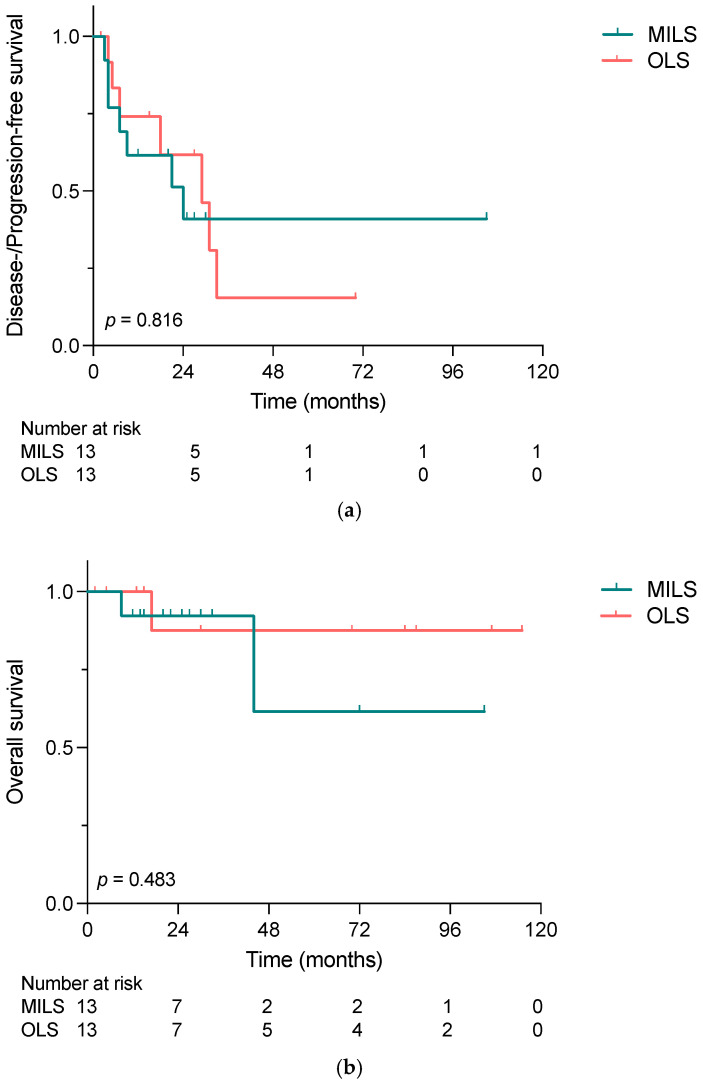
(**a**) Disease-/progression-free survival and (**b**) overall survival of patients who underwent liver-directed surgery for NELM comparing minimally invasive and open liver surgery, calculated with the Kaplan–Meier method. Neither DFS/PFS nor OS differed significantly between MILS and OLS (*p* = 0.816 and *p* = 0.483) groups. MILS, minimally invasive liver surgery; OLS, open liver surgery.

**Table 1 cancers-15-02048-t001:** Characteristics of the patients who underwent liver-directed surgery for NELM.

Gender ^1^	Female Male	30 (68%) 22 (42%)
Age (years) ^2^		60 (21–80)
BMI (kg/m^2^) ^2^		25 (19–36)
ASA ^1^	2 3	32 (62%) 20 (38%)
Localization of primary ^1^	Pancreas Small intestine	23 (44%) 29 (56%)
Resection of Primary ^1^		51 (98%)
Grading of Primary ^1^	G1 G2 G3	18 (35%) 33 (63%) 1 (2%)
Appearance of metastases ^1^	Synchronous Metachronous	38 (73%) 14 (27%)
Indication ^1^	Curative Debulking	22 (42%) 30 (58%)
Technique ^1^	Open Minimally invasive	39 (75%) 13 (25%)
Duration of surgery (minutes) ^2^		267 (90–575)
Extent of surgery ^1^	Major resection Minor resection	17 (33%) 35 (67%)
Simultaneous surgery of primary		23 (44%)
ICU ^1^		44 (85%)
Length of ICU stay ^2^		1 (0–35)
Length of hospital stay ^2^		11 (6–132)
90-day complications ^1^		24 (46%)
90-day major complications ^1^		17 (32%)
90-day mortality ^1^		0 (0%)
Grading of hepatic metastases ^1^	G1 G2 G3	19 (36%) 28 (54%) 5 (10%)
R status ^1^	R0 R1 R2	21 (40%) 1 (2%) 30 (58%)
Neoadjuvant therapy ^1^		22 (42%)
Adjuvant therapy ^1^		41 (79%)
>1 adjuvant therapy modalities ^1^		26 (50%)
>2 adjuvant therapy modalities ^1^		15 (29%)

^1^ Count (percentage); ^2^ Median (range); NELM, neuroendocrine liver metastases; BMI, Body Mass Index; ASA, American Society of Anesthesiologists; ICU, Intensive Care Unit.

**Table 2 cancers-15-02048-t002:** Characteristics of the patients who underwent liver-directed surgery for NELM according to the indication for surgery.

		Curative (*n* = 22)	Debulking (*n* = 30)	*p*
Gender ^1^	Female Male	16 (73%) 6 (27%)	14 (47%) 16 (53%)	0.060
Age (years) ^2^		63 (31–80)	59 (21–75)	0.270
BMI (kg/m^2^) ^2^		25 (20–33)	26 (19–36)	0.977
ASA ^1^	1 2 3	0 (0%) 15 (68%) 7 (32%)	0 (0%) 16 (53%) 14 (47%)	0.399
Localization of primary ^1^	Pancreas Small intestine	8 (36%) 14 (64%)	15 (50%) 15 (50%)	0.328
Resection of Primary ^1^		22 (100%)	29 (97%)	0.387
Grading of Primary ^1^	G1 G2 G3	8 (36%) 14 (64%) 0	10 (33%) 19 (64%) 1 (3%)	0.681
Appearance of metastases ^1^	Synchronous Metachronous	15 (68%) 7 (32%)	23 (77%) 7 (23%)	0.496
Technique ^1^	Open Minimally invasive	17 (77%) 5 (23%)	22 (73%) 8 (27%)	0.746
Duration of surgery (minutes) ^2^		251 (90–575)	288 (130–499)	0.476
Extent of surgery ^1^	Major resection Minor resection	8 (37%) 14 (64%)	9 (30%) 21 (70%)	0.483
Simultaneous surgery of primary ^1^		7 (32%)	16 (53%)	0.123
ICU ^1^		18 (82%)	26 (87%)	0.632
Length of ICU stay ^2^		2 (0–8)	1 (0–35)	0.503
Length of hospital stay ^2^		11 (6–42)	12 (6–132)	0.558
90-day complications ^1^		10 (45%)	14 (47%)	0.931
90-day major complications ^1^		7 (32%)	10 (33%)	0.908
90-day mortality ^1^		0 (0%)	0 (0%)	1.0
Grading of hepatic metastases ^1^	G1 G2 G3	9 (41%%) 10 (45%) 3 (14%)	10 (33%) 18 (60%) 2 (7%)	0.512
R status ^1^	R0 R1 R2	21 (96%) 1 (4%) 0 (0%)	0 (0%) 0 (0%) 30 (100%)	<0.001
Neoadjuvant therapy ^1^		6 (27%)	16 (53%)	0.049
Adjuvant therapy ^1^		16 (73%)	25 (83%)	0.147
>1 adjuvant therapy modalities ^1^		7 (32%)	19 (63%)	0.025
>2 adjuvant therapy modalities ^1^		4 (18%)	11 (37%)	0.146

^1^ Count (percentage); ^2^ Median (range); NELM, neuroendocrine liver metastases; BMI, Body Mass Index; ASA, American Society of Anesthesiologists; ICU, Intensive Care Unit.

**Table 3 cancers-15-02048-t003:** Characteristics of patients undergoing liver-directed surgery according to the surgical approach after propensity score matching.

		MILS (*n* = 13)	OLS (*n* = 13)	*p*
Gender ^1^	Female Male	6 (46%) 7 (54%)	6 (46%) 7 (54%)	1.0
Age (years) ^2^		57 (46–73)	61 (31–76)	0.960
BMI (kg/m^2^) ^2^		25 (20–36)	24 (19–29)	0.801
ASA ^1^	2 3	5 (38%) 8 (62%)	6 (46%) 7 (54%)	0.691
Localization of primary ^1^	Pancreas Small intestine	6 (46%) 7 (54%)	5 (38%) 8 (62%)	0.691
Appearance of metastases ^1^	Synchronous Metachronous	10 (77%) 3 (23%)	8 (62%) 5 (38%)	0.395
Indication ^1^	Curative Debulking	5 (38%) 8 (62%)	5 (38%) 8 (62%)	1.0
Duration of surgery (minutes) ^2^		285 (130–504)	245 (209–431)	0.880
Extent of surgery ^1^	Major resection Minor resection	3 (23%) 10 (77%)	5 (38%) 8 (62%)	0.395
Simultaneous resection of primary ^1^		4 (31%)	3 (23%)	0.658
ICU ^1^		11 (85%)	12 (92%)	0.539
Length of ICU stay ^2^		1 (0–6)	1 (0–35)	0.579
Length of hospital stay ^2^		10 (7–20)	14 (9–87)	0.034
90-day complications ^1^		6 (46%)	6 (46%)	1.0
90-day major complications ^1^		5 (38%)	4 (31%)	0.680
90-day mortality ^1^		0 (0%)	0 (0%)	1.0
Grading Hepatic Metastases ^1^	G1 G2 G3	4 (31%) 8 (61%) 1 (8%)	4 (31%) 8 (61%) 1 (8%)	1.0
R status ^1^	R0 R1 R2	4 (31%) 1 (8%) 8 (61%)	5 (38%) 0 (0%) 8 (62%)	0.574
Resection of Primary ^1^		12 (92%)	13 (100%)	0.308
Grading Primary ^1^	G1 G2 G3	4 (31%) 8 (61%) 1 (8%)	4 (31%) 9 (69%) 0 (0%)	0.589
Neoadjuvant therapy ^1^		3 (23%)	4 (31%)	0.658
Adjuvant therapy ^1^		11 (85%)	11 (85%)	1.0
>1 adjuvant therapy modalities ^1^		6 (46%)	8 (61%)	0.431
>2 adjuvant therapy modalities ^1^		3 (23%)	4 (31%)	0.658

^1^ Count (percentage); ^2^ Median (range); MILS, Minimally invasive liver surgery; OLS, Open liver surgery; BMI, Body Mass Index; ASA, American Society of Anesthesiologists; ICU, Intensive Care Unit.

## Data Availability

Not applicable.

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
