# Peer review of "Surgical Approach to Liver Metastases in GEP-NET in a Tertiary Reference Center"

_cancers, 2023, doi:10.3390/cancers15072048_

Round 1

Reviewer 1 Report

The article presents the results of a retrospective cohort study about hepatic resection for hepatic metastases of GEP-NET. The authors found that oncological outcomes could be achieved via curative intended as well as debulking surgery and minimally invasive liver surgery seems to be not inferior to open one concerning survival rates.

Neuroendocrine tumors are a heterogenous group of relatively rare neoplasms. The role of debulking surgery for neuroendocrine liver metastases is still controversial. Further research is needed in this field.

The manuscript is well written and well structured. The text is clear and easy to read. The topic is interesting and in line with the journal.

In my opinion, the article can be published in its current form.

Reviewer 2 Report

Comments to authors:

In their paper entitled “Surgical approach to liver metastases in GEP-NET in a tertiary reference center”, Butz et al. describe their institutional experience with surgical treatment of NELM over 12 years. This is a commendable effort for a rare disease. However, the small sample size and heterogeneous population in this study (including different intent of surgery, location of primary tumor, and tumor grades) preclude any safe conclusions in this paper.

Specific comments:

1)      I am not sure that including biopsies (“tissue sampling surgery”) in the same group as curative/debulking surgeries makes a lot of sense. As noted in Table 2, everything is different about these cases. Including these makes the results more heterogenous and difficult to interpret.

2)      The statistical power of the survival analyses is very low due to the small sample size, especially after stratifying by intent, tumor grade, MIS, etc. This should be highlighted in the limitations and the conclusions of these comparisons should be more moderate (i.e. instead of claiming there was no difference, consider a statement acknowledging that the small sample size would not allow for safe conclusions).

3)      Similar to above – the sample size is too small for any meaningful comparative analyses. This should either be highlighted or these analyses should be removed.

4)      The entire discussion is based on the comparative outcomes analyses, which, as I noted above, do not hold much weight in light of the small sample size.

Minor comments:

1)      Intro, first paragraph. While there could be some regional bias on this, I do not think that the role of debulking surgery for NELM is controversial anymore (see Ann Surg Oncol. 2020 Sep;27(9):3270-3280 and NCCN guidelines).

2)      Methods, last paragraph. I believe the authors mean that significance was set at 0.05 rather than 0.5?

Reviewer 3 Report

The authors performed a review of patients who underwent hepatic surgery for metastatic enteropancreatic neuroendocrine tumors between 2010 and December 2022. The first striking feature of the study is that the patient inclusion period ended less than 2 months ago. This is completely unacceptable in a study analyzing 90-day complications or survival.

Secondly, it is noteworthy that patients who underwent tissue sampling were included. These patients are not comparable to the other resections and should therefore be excluded.

Thirdly, it is surprising that patients who underwent debulking surgery or biopsies have an R0 or R1 status. This is quite contradictory, as it would suggest that the resection has been complete and therefore curative. Having clarified this, it would be interesting to know the % of patients operated on with curative intent in both groups in whom an R0 resection was obtained.

Fourth, progression-free survival is used indiscriminately in patients with and without residual disease. It is not the same as disease-free survival, so it should be differentiated in R0 tumors, which has been the % of recurrence in the different groups and the disease-free survival in each of them. In any case, it is not appropriate to express % survival at 5 years when the median follow-up is much less than 5 years.

Fifthly, the authors do not consider synchronous surgery of the primary tumor in the propensity score matching. However, in the unpaired data this is a differential feature between the two groups and after matching it remains at the limit of significance, due to the small sample size. This is a critical issue as it is likely to have a very important influence on postoperative outcomes. All the differences observed and attributed to the use of minimally invasive approaches could be justified by this aspect, so it is essential to include it in the matching model.

Finally, given the small sample size of the study, with a discrete follow-up and all the limitations mentioned, the authors should be less categorical in their assertions in the discussion, and avoid terms such as " demonstrated."

The text of section 2.1 "clincal" and the significance level of "0.5" in the statistical analysis section should be revised.

Round 2

Reviewer 2 Report

This revised version addressed the limitations of the original version satisfactorily. While the low sample size remains, the analyses and discussion acknowledge this and are in my opinion appropriate.

Reviewer 3 Report

Congratulations to the authors for the revision of the manuscript, which has significantly improved its quality.